

# Clinical diagnostic value of amino acids in laryngeal squamous cell carcinomas

Shousen Hu[1,*], Chang Zhao[2,*], Zi'an Wang[3,4], Zeyun Li[3,4] and Xiangzhen Kong[3,4]

[1] Department of Otolaryngology Head and Neck Surgery, The First Affiliated Hospital of Zhengzhou University, Zhengzhou, China
[2] Department of Otorhinolaryngology, Zhengzhou Seventh People's Hospital, Zhengzhou, China
[3] Department of Pharmacy, The First Affiliated Hospital of Zhengzhou University, Zhengzhou, China
[4] Henan Key Laboratory of Precision Clinical Pharmacy, Zhengzhou University, Zhengzhou, China
* These authors contributed equally to this work.

Corresponding author
Xiangzhen Kong,
kongxiangzhen118@126.com

## ABSTRACT

**Background:** Early diagnosis and treatment can improve the survival rates of patients with laryngeal squamous cell carcinoma (LSCC). Therefore, it is necessary to discover new biomarkers for laryngeal cancer screening and early diagnosis.

**Methods:** We collected fasting plasma from LSCC patients and healthy volunteers, as well as cancer and para-carcinoma tissues from LSCC patients, and performed quantitative detection of amino acid levels using liquid chromatography-mass spectrometry. We used overall analysis and multivariate statistical analysis to screen out the statistically significant differential amino acids in the plasma and tissue samples, conducted receiver operating characteristic (ROC) analysis of the differential amino acids to evaluate the sensitivity and specificity of the differential amino acids, and finally determined the diagnostic value of amino acids for laryngeal cancer. Additionally, we identified amino acids in the plasma and tissue samples that are valuable for the early diagnosis of laryngeal cancer classified according to the tumor-node-metastasis (TNM) classification.

**Results:** Asparagine (Asp) and homocysteine (Hcy) were two amino acids of common significance in plasma and tissue samples, and their specificity and sensitivity analysis showed that they may be new biomarkers for the diagnosis and treatment of LSCC. According to the TNM staging system, phenylalanine (Phe) and isoleucine (Ile) were screened out in the plasma of LSCC patients at early (I and II) and advanced (III and IV) stages; ornithine hydrochloride (Orn), glutamic acid (Glu), and Glycine (Gly) were selected in the tissue. These dysregulated amino acids found in LSCC patients may be useful as clinical biomarkers for the early diagnosis and screening of LSCC.

## INTRODUCTION

Laryngeal squamous cell carcinoma (LSCC) is one of the most common head and neck tumors, accounting for more than 90% of laryngeal cancer cases (*Megwalu & Sikora,*

*2014*). In the last 40 years, the incidence of LSCC has increased significantly, and most of the patients are 40–60 years old with a male to female ratio of (7–9):1 (*Li, 2014*; *Siegel, Miller & Jemal, 2019*). Some patients are diagnosed with distant metastasis or advanced stage laryngeal cancer due to untimely diagnosis or treatment (*Smith, Abrol & Gardner, 2016*), which could lead to unsatisfactory treatment results and serious psychological and economic burdens to patients (*Guibert et al., 2011*). Therefore, the early identification of laryngeal cancer is very necessary. Although some current imaging methods (such as magnetic resonance imaging, positron emission computed tomography, and computed tomography) are frequently used to screen and diagnose laryngeal cancer, their effectiveness depends on the ability of medical imaging workers to obtain image information (*Yu & Wang, 2021*). Hence, it is highly important to find more effective, simple, and less invasive diagnostic techniques that can be used for laryngeal cancer screening or early detection.

A possible way to improve the screening or early detection of LSCC is the use of biomarkers. LysoPC 16:0 and PAF might be diagnostic biomarkers for laryngeal cancer based on serum metabolomics from patients and healthy individuals (*Zhang et al., 2018*). Six metabolites (D-pantothenic acid, palmitic acid, myristic acid, oleamide, sphinganine, and phytosphingosine) could be used as diagnostic biomarkers for laryngeal cancer according to urine metabolomic data (*Chen et al., 2019*). Lysophospholipids and phospholipids might serve as lipid biomarkers for the diagnosis of laryngeal cancer (*Yu & Wang, 2021*). Amino acid metabolomics is an effective method for identifying biomarkers of disease-related metabolites that contribute to the early diagnosis of diseases (*Siminska & Koba, 2016*). It is widely used in lung cancer (*Maeda et al., 2010*), gastric cancer (*Li et al., 2022*), colorectal cancer (*Leichtle et al., 2012*), head and neck cancer (*Cadoni et al., 2020*), and multiple myeloma (*Yue et al., 2022*). Currently, there are three main kinds of amino acid detection methods: high performance liquid chromatography (HPLC), amino acid analysis, and liquid chromatography-mass spectrometry (LC-MS). The poor resolution of amino acids and the difficult separation of isomers make the results of HPLC unreliable. Amino acid analyzers take a long time to analyze blood samples and cannot be used as a high-throughput analysis method. Due to its high-efficiency separation ability, high sensitivity, high accuracy, and short detection time, LC-MS has become the preferred detection method for metabolomics (*Theodoridis et al., 2012*). As far as we know, no laryngeal cancer-associate amino acid profiling of blood and tissue samples has yet been performed.

In order to find reliable clinical biomarkers for laryngeal cancer screening, we used LC-MS to analyze the amino acid levels in the plasma of 20 LSCC patients and 15 healthy volunteers, as well as LSCC patients' cancer tissues and adjacent tissues. The different amino acids in the plasma and tissue samples were determined and amino acids with diagnostic values were screened out. Our study provides new strategies for identifying reliable clinical biomarkers for the early diagnosis and screening of LSCC.

The tumor-node-metastasis (TNM) staging system based on anatomical study contributes to cancer treatment selection, control activities, and outcome prediction (*Huang & O'Sullivan, 2017*). According to the eighth edition of the TNM staging system,

we performed TNM staging for LSCC patients and analyzed the differential amino acids in the plasma and tissues of LSCC patients at early (I and II) and advanced (III and IV) stages. Our study provides further evidence for the early prognostic prediction of LSCC.

## MATERIALS AND METHODS

### Clinical subjects

This study was approved by the Ethics Committee of the First Affiliated Hospital of Zhengzhou University (2021-KY-0417-002). Tissue samples and blood samples were obtained from patients admitted to the First Affiliated Hospital of Zhengzhou University from October 2021 to February 2022. In this study, fasting blood samples were collected from 20 LSCC patients and 15 healthy individuals (with no past or current history of cancers), as well as laryngeal cancer tissues and para-carcinoma tissues from 20 LSCC patients (one patient with missing para-carcinoma tissue). The inclusion criteria for this study were as follows: a diagnosis of LSCC was reported by pathological examination, with or without cervical lymph node metastasis. The exclusion criteria were: (1) patients who had previously or currently suffered from other tumors; (2) patients who had received chemotherapy or radiotherapy prior to surgery; (3) patients with severe blood, metabolic, respiratory, or other system diseases; and (4) patients suffering from infectious diseases such as hepatitis B, syphilis, and human immunodeficiency virus.

### Blood sample preparation and metabolite extraction

All selected subjects were required to fast for more than 8 h before blood collection, and blood from LSCC patients was collected before surgery. The blood samples were placed in a vacuum anticoagulant tube containing lithium heparin, centrifuged at 4 °C, 3,000 rpm for 10 min, and centrifugation was completed within 1 h after the blood samples were collected. The plasma was stored at −80 °C. An appropriate amount of plasma samples were pipetted accurately into 2 mL EP tubes, 400 μL of 10% formic acid methanol-$H_2O$ (1:1, V/V) solution was added, and then centrifuged at 12,000 rpm, 4 °C for 5 min. Following that, 10 μL of the supernatant was taken and added to 90 μL of 10% formic acid methanol-$H_2O$ (1:1, V/V) solution. We took 100 μL of the diluted sample and added it to 100 μL of the isotope internal standard with a concentration of 100 ppb. After being vortexed for 30 s, the supernatant was filtered through a 0.22 μm membrane, and collected in a detection bottle. Content (μg/mL) = C (ng/mL) * 0.5 * 2 * 10/sampling amount (μL).

### Tissue sample preparation and metabolite extraction

The LSCC tissue and para-carcinoma tissue of patients were collected during the operation, immediately put into a liquid nitrogen tank, and then stored at −80 °C. An appropriate amount of tissue samples were pipetted accurately into 2 mL EP tubes, added to 600 μL of 10% formic acid methanol solution-$H_2O$ (1:1, V/V) solution, and two steel balls were added, put into a tissue grinder, 60 Hz grinding for 90 s, and then centrifuged at 12,000 rpm, 4 °C for 5 min. We took 10 μL of the supernatant and added it to 190 μL of 10% formic acid methanol-$H_2O$ (1:1, V/V) solution. The remaining sample

**Table 1 Clinical characteristics of the subjects for amino acids metabolism analysis.**

| Characteristics | Serum sample | | Tissue sample | |
|---|---|---|---|---|
| | LSCC | NC | LSCC | ParaLC |
| No. | 20 | 15 | 20 | 19 |
| Sex (male/female) | 20/0 | 15/0 | 20/0 | 19/0 |
| Age (years) | 62.75 ± 7.83 | 58.47 ± 4.67 | 62.8 ± 10.04 | 63.63 ± 9.58 |
| TNM stage (AJCC 8) | | | | |
| Stage I+II | 12 | / | 10 | / |
| Stage III+IV | 8 | / | 10 | / |

Note:
    AJCC, American joint committee on cancer; T, tumor; N, node; M, metastasis.

pretreatment steps were the same as blood sample. Diluted 20-fold content (µg/g) = C (ng/mL) * 0.6 * 2 * 20/sampling amount (mg).

## LC-MS analysis

Amino acid metabolism was performed on Waters ACQUITY UPLC (Waters, Milford, MA, USA) coupled with the AB5000 mass spectrometer (AB SCIEX, Framingham, MA, USA) system. We separated 5 µL of the diluted sample on the ACQUITY UPLC®BEH C18 column (2.1 × 100 mm, 1.7 µm; Waters, Milford, MA, USA). The LC-MS conditions were mainly optimized on the basis of the literature (*Liyanaarachchi et al., 2018*; *Thiele et al., 2019*). The column temperature was maintained at 40 °C. The mobile phase A in HPLC consisted of 10% methanol water, and the mobile phase B was 50% methanol water, both containing 0.1% formic acid. The gradient profile was as follows: 10–30% B in 0–6.5 min; 30–100% B in 6.5–7 min; 100% B in 7–14 min; and 100–10% B in 14–17.5 min. In MS detection, a multiple reaction monitoring mode was used with an electronic spray ion source operating in positive ion mode. The ion source temperature was 500 °C and the spray voltage was 5,500 V. The collision gas was 6 psi, curtain gas was 30 psi, and both the atomizing gas and auxiliary gas were 50 psi.

## Statistical analysis

Unit variance scaling was performed to normalize the data before multivariate statistical analysis for more reliable and intuitive results. Partial least squares discriminant analysis (PLS-DA) was conducted by the Ropls package in R to distinguish different groups. The heatmap, box plots and receiver operating characteristic (ROC) curves were performed using the pheatmap, ggpubr and pROC packages in R, respectively. Nonparametric Mann-Whitney test was conducted to identify the levels of differential amino acids between the LSCC and NC/ParaLSCC groups.

## RESULTS

### Clinical characteristics of the subjects

The detailed clinical characteristics of the subjects were summarized in Table 1. LSCC was classified according to the 8th Edition American Joint Committee on Cancer (AJCC)

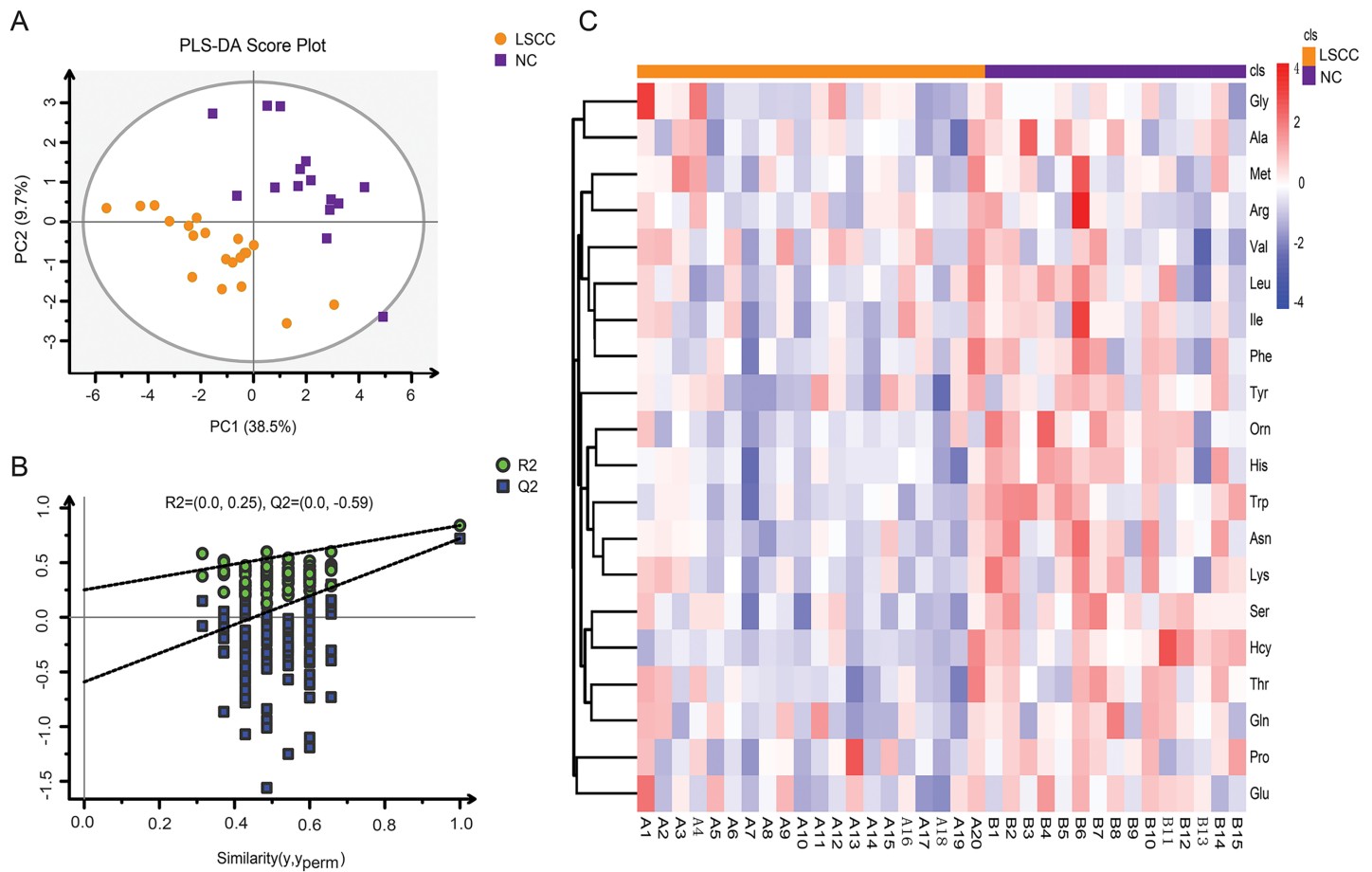

**Figure 1** **Amino acid profiling of the plasma samples from LSCC patients and NC subjects.** (A) Partial least squares discriminant analysis (PLS-DA) scores between LSCC and NC groups in plasma samples. PC1 and PC2 were the first and second principal components, respectively, and the percentage values represented the proportion of the total variance by each component. (B) The permutation test of PLS-DA model between LSCC and NC groups ($n = 100$). (C) Overall heatmap of amino acid metabolites in plasma samples. Each section's colour represents the significance of changes in metabolites (red, up-regulated; blue, down-regulated).

Staging Manual (*Amin et al., 2017*). There was no statistical difference in age between the LSCC and NC groups, and both included only male participants.

## Amino acid signatures of plasma and tissue samples from laryngeal carcinoma

Amino acid profiling was performed to analyze the plasma and tissue samples from LSCC patients and NC subjects. The levels of amino acids in the plasma and tissue samples are shown in Tables S1 and S2. PLS-DA model was used to explore the differences in amino acid fingerprinting between LSCC patients and NC subjects in plasma samples. Both the $R^2$ and $Q^2$ of the established PLS-DA model were greater than 0.5, indicating that this model had good explanatory and predictive ability. The scores are shown in Fig. 1A, and we found that the LSCC group plasma samples were significantly different from those of the NC group. The reliability of the PLS-DA model was further verified by the permutation test ($n = 100$). The $R^2$ cutoff was 0.25 and the $Q^2$ cutoff was −0.59 in the permutation test,

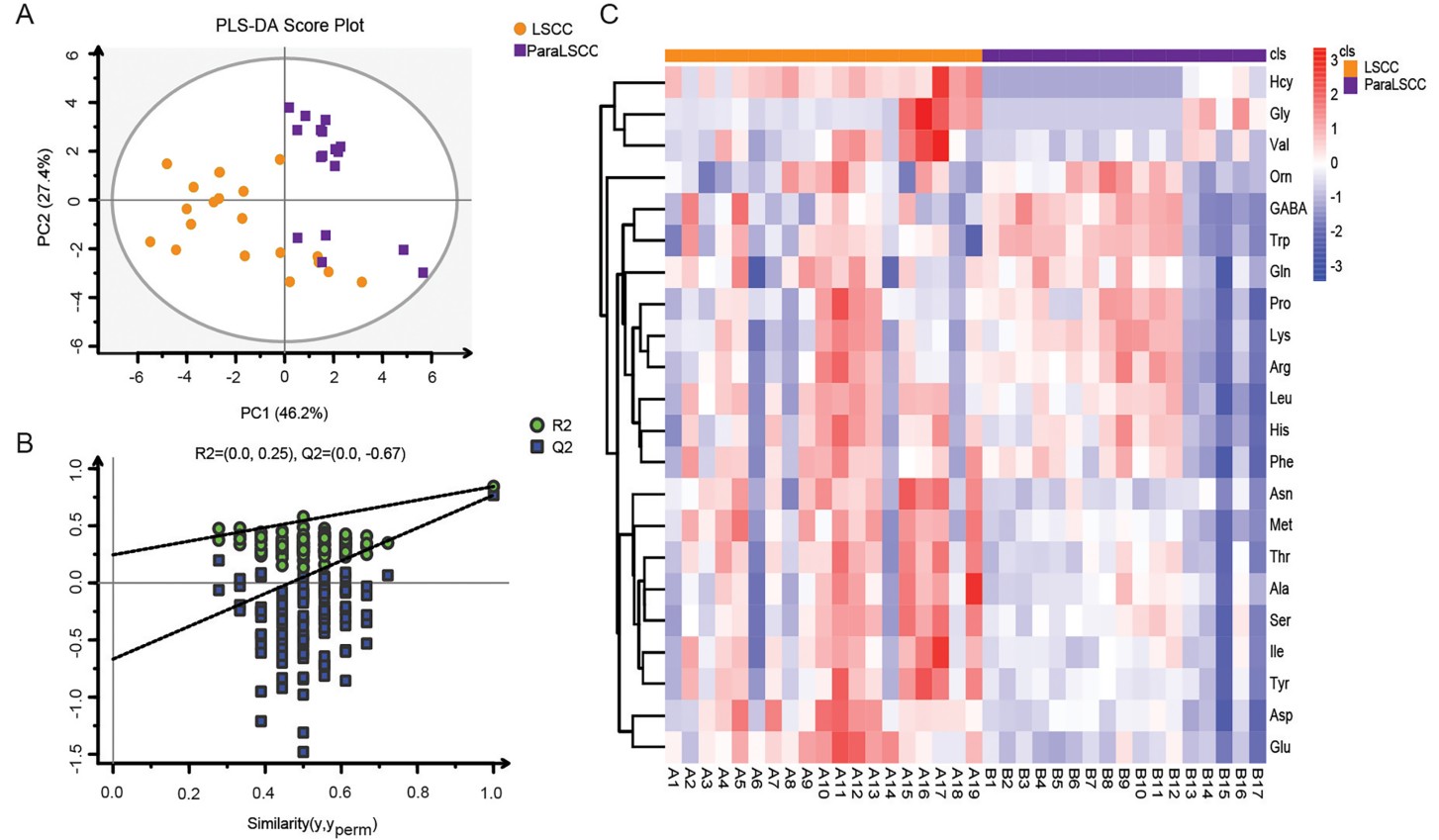

**Figure 2** **Amino acid profiling of the LSCC and ParaLSCC tissues from LSCC patients.** (A) PLS-DA scores and the permutation test of PLS-DA model between the LSCC and ParaLSCC tissues. (B) The permutation test of PLS-DA model between the LSCC and ParaLSCC tissues ($n = 100$). (C) Overall heatmap of amino acid metabolites in tissue samples. Each section's colour represents the significance of changes in metabolites (red, up-regulated; blue, down-regulated).

indicating that this model was not over-fitting (Fig. 1B). Heatmap visualized the alterations of amino acids in LSCC patients and NC subjects (Fig. 1C). The amino acid signatures of tissue samples from laryngeal carcinoma are shown in Fig. 2.

## Potential biomarkers of amino acids in laryngeal cancer

Variable importance in projection (VIP) is the variable weight value of the PLS-DA model, and can be used to measure the impact strength and explanatory power of the differences in the accumulation of each metabolite on the classification and discrimination of each group of samples (*Liu et al., 2021*). VIP ≥ 1 was considered as one of the screening standards for potential biomarkers. Combined with the standard of VIP >1 in the PLS-DA model and $P < 0.05$ in *t*-test (Tables S3 and S4), six differential amino acids were finally screened out in the plasma, namely asparagine (Asn), serine (Ser), homocysteine (Hcy), histidine (His), ornithine hydrochloride (Orn) and tryptophan (Trp); eight differential amino acids were screened in the tissue, namely tyrosine (Tyr), Asn, isoleucine (Ile), methionine (Met), threonine (Thr), aspartic acid (Asp), glutamic acid (Glu), and Hcy. There were two amino acids of common significance in plasma and tissue samples, Asn and Hcy (Fig. 3). These two amino acids may have important significance for the early

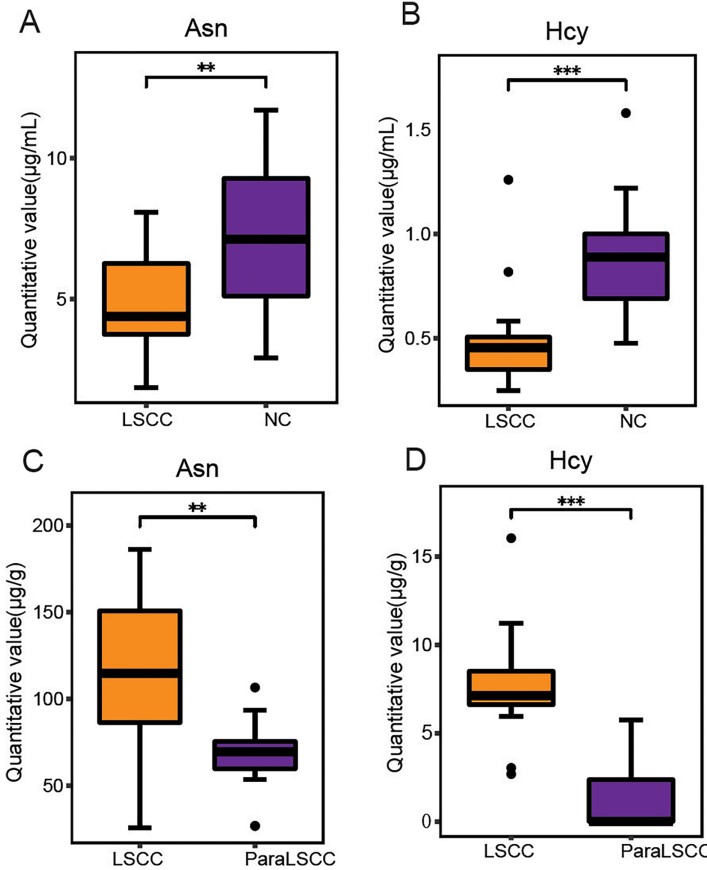

**Figure 3 Commonly different amino acids in plasma and tissue samples.** The quantitative value of Asn and Hcy in plasma samples of LSCC and NC groups (A and B), and in the LSCC and ParaLSCC tissues from LSCC patients (C and D). **$P < 0.01$, ***$P < 0.001$.

clinical diagnosis of LSCC. Receiver operating characteristic (ROC) analysis was employed to evaluate the diagnostic performance of these two biomarkers for LSCC. In plasma samples, the area under curve (AUC) values of Hcy and Asn were 0.765 and 0.896, respectively (Figs. 4A and 4B); in tissue samples, the AUC values of Hcy and Asn were 0.807 and 0.980, respectively (Figs. 4D and 4E). When Hcy and Asn were chosen as the 'combined biomarker', the AUC value was 0.833 in plasma samples, and 1.000 in tissue samples (Figs. 4C and 4F).

## Amino acid profiling between stage I+II patients and stage III+IV patients according to the TNM staging system

LSCC patients were divided into stage I+II and stage III+IV patients according to the eighth edition of TNM staging system. The PLS-DA model was used to identify differential levels of amino acids in plasma and tissue samples between these two groups. In Figs. 5A and 5B, a clear distinction between LSCC stage I+II and stage III+IV patients was determined in the plasma and tissue samples. Heatmap visualized the alterations of amino acids in LSCC stage I+II and stage III+IV patients (Figs. 5C and 5D). Combined with the

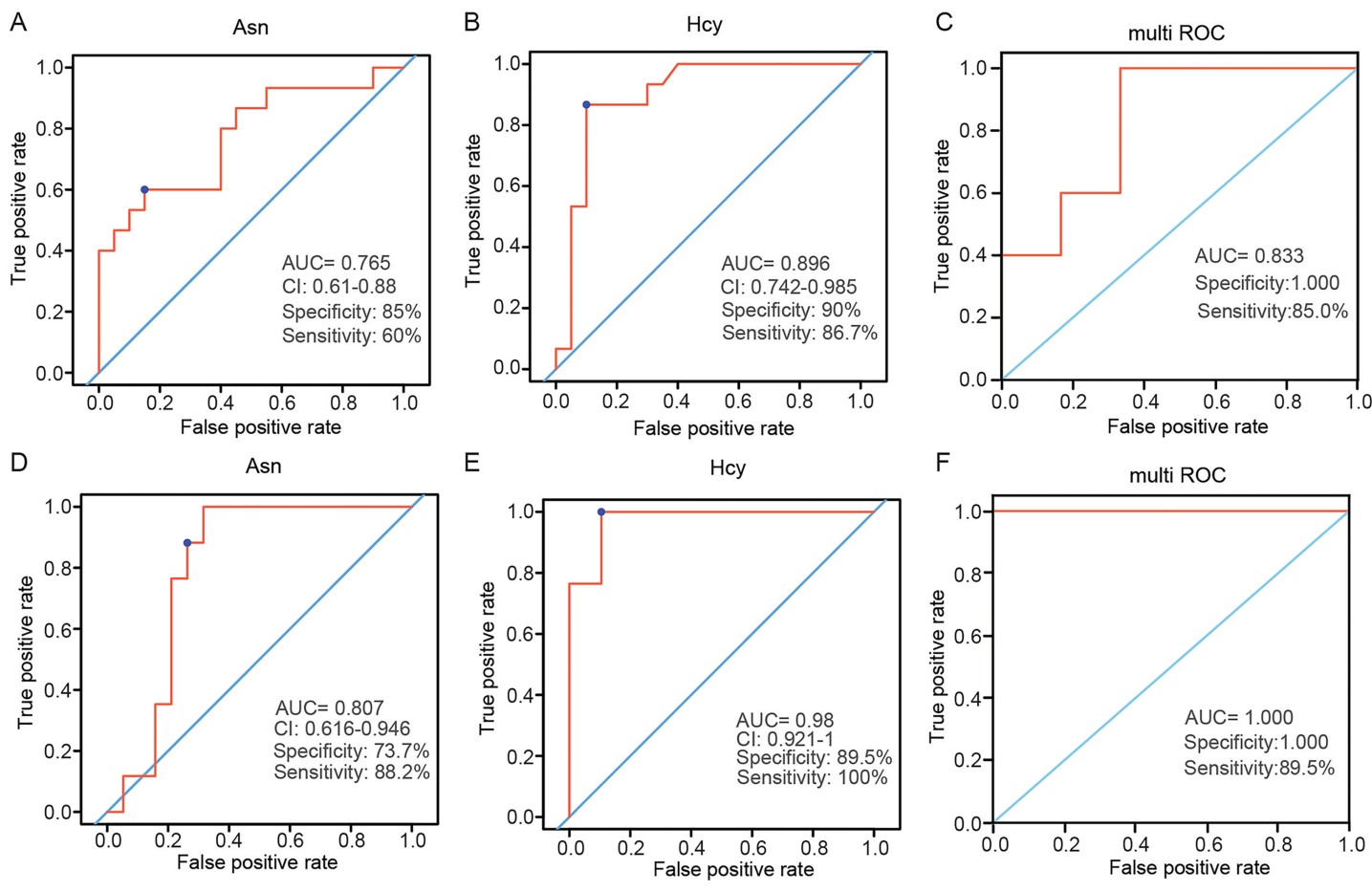

**Figure 4** **Receiver operating characteristic (ROC) analysis of two commonly different amino acids in plasma and tissue samples.** ROC analysis of Asn and Hcy, and their combination analysis in plasma samples (A–C) and in tissue samples (D–F). AUC, area under curve.

standard of VIP > 1 in the PLS-DA model and $P < 0.05$ in the $t$-test (Tables S5 and S6), two differential amino acids were finally screened out in the plasma, namely Phe and Ile; three differential amino acids were selected out in the tissue, namely Orn, Glu and Gly.

In plasma samples, the AUC of Phe and Ile were 0.781 and 0.792, respectively (Figs. 6A and 6B); in tissue samples, the AUC of Orn, Glu, and Gly were 0.85, 0.867, and 0.944, respectively (Figs. 6C–6E).

## DISCUSSION

Researchers have paid increasingly more attention to the important roles of amino acid metabolomics in tumor pathogenesis (*Sivanand & Vander Heiden, 2020*). However, there have been few reports on the amino acid metabolomics of LSCC. In this study, we collected the fasting plasma of LSCC patients and healthy volunteers, cancer and para-carcinoma tissues of LSCC patients, and used LC-MS to quantitaively determine the levels of amino acids in plasma and tissue samples. The different amino acids were screened out and the amino acids with high sensitivity and specificity were identified as new biomarkers for LSCC. In the plasma samples, we found six amino acids with statistical differences, which

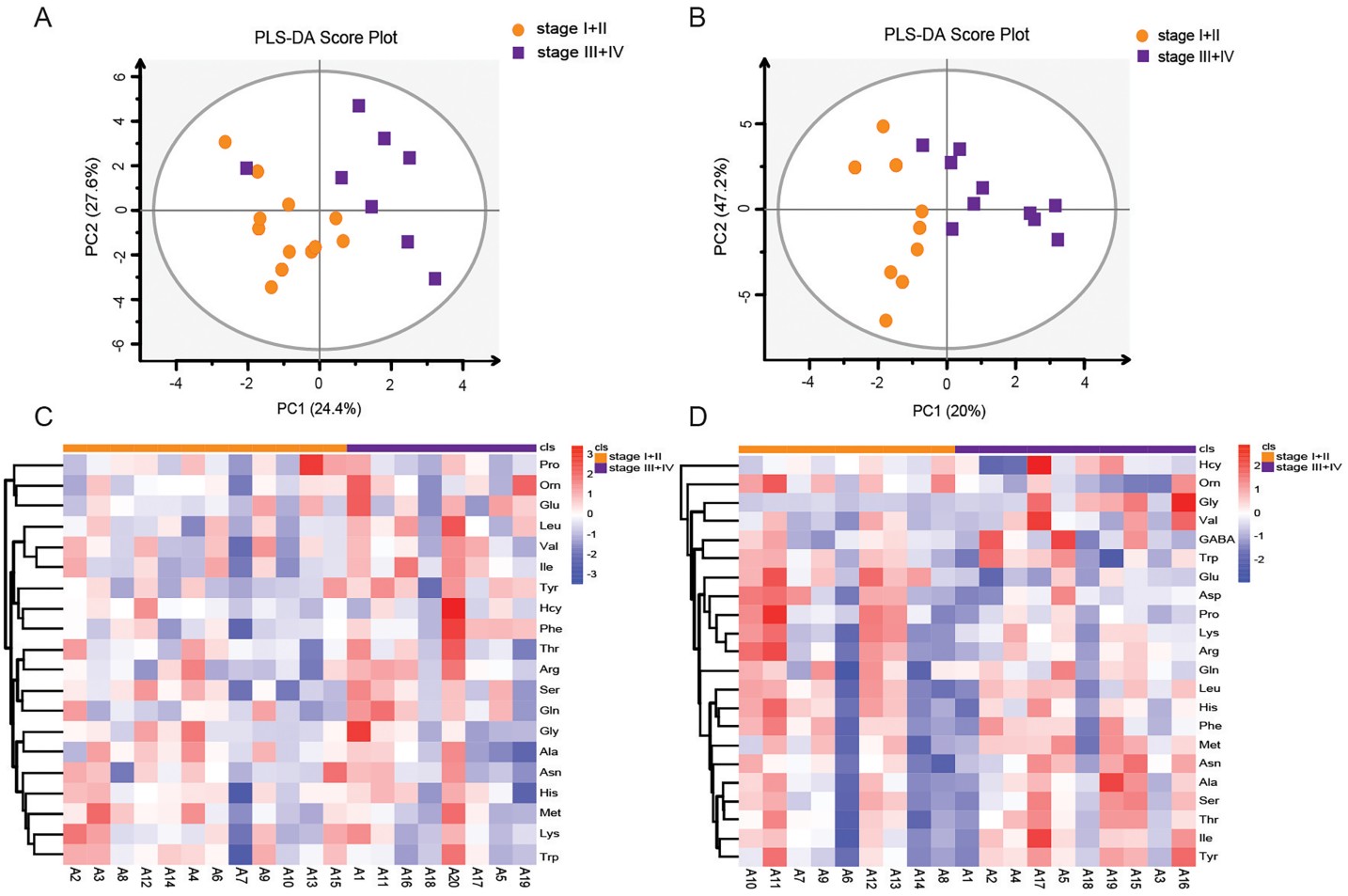

**Figure 5 Amino acid profiling of the plasma and tissue samples between stage I+II and stage III+IV patients according to the TNM staging system.** PLS-DA scores of amino acids in plasma (A) and tissue (B) samples between LSCC stage I+II and stage III+IV patients. Overall heatmap of amino acids in plasma (C) and tissue (D) samples between LSCC stage I+II and stage III+IV patients.

were Asn, Ser, Hcy, His, Orn and Trp; in tissue samples, eight amino acids were found with statistical differences, which were Tyr, Asn, Ile, Met, Thr, Asp, Glu, and Hcy. In both plasma and tissue samples, Hcy and Asn were statistically significant amino acids, and were more valuable as a combined biomarker than individual markers. In addition, differential levels of amino acids in plasma and tissue samples between LSCC stage I+II and stage III+IV patients were screened out. Two differential amino acids were screened out in the plasma, namely Phe and Ile; three differential amino acids were screened out in the tissue, namely Orn, Glu and Gly.

Hcy level is one of the key factors associated with cancers (*Hasan et al., 2019*) such as colorectal (*Kato et al., 1999*) and cervical cancer (*Silva et al., 2020*). Hcy can directly damage DNA by generating reactive oxygen species, which may be carcinogenic (*Oikawa, Murakami & Kawanishi, 2003*; *Zhou et al., 2016*). In this study, Hcy was one differential amino acid found in plasma and tissue samples. In the plasma samples, the level of Hcy in the LSCC patient group was significantly lower than that in the healthy control group ($P < 0.001$), with an AUC of 0.896, specificity of 0.9, and sensitivity of 0.867. The level of

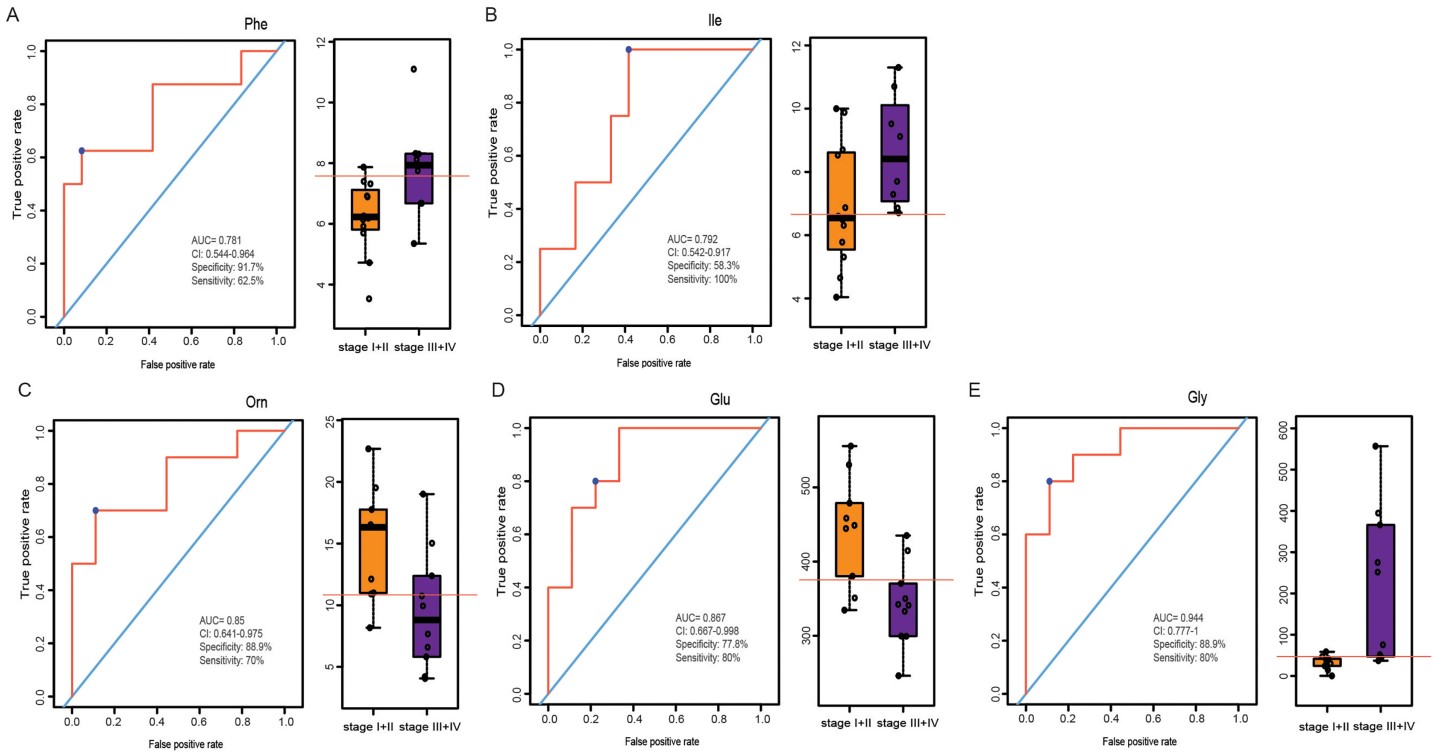

**Figure 6 ROC analysis of the different amino acids in plasma and tissue samples between stage I+II and stage III+IV patients.** ROC analysis of Phe and Ile in plasma samples (A–B), and Orn, Glu and Gly in tissue samples (C–E).

Hcy in LSCC tissue was significantly higher than in the adjacent tissue ($P < 0.001$), with an AUC was 0.98, specificity of 0.895, and sensitivity of 1. Ultimately, Hcy was the most significant differential amino acid in this study. Hcy may be closely related to the pathogenesis of LSCC and may be a new biomarker for diagnosis of LSCC in the future.

Asn, a non-essential amino acid, plays an important role in cellular metabolism, and in tumor cells, the extracellular source of Asn is critical for cell proliferation (*Figueiredo, Cole & Drachtman, 2016*; *Wang et al., 2016*). Asn was significantly altered in patients with non-small cell lung cancer (*Liu et al., 2022*). Inhibition of Asn can trigger the protective autophagy of cancer cells (*Ji et al., 2017*). In this study, the differential level of Asn in plasma and tissue samples was statistically significant. In plasma samples, the level of Asn in the LSCC patient group was lower than that of healthy people, with an AUC of 0.765, specificity of 0.85, and sensitivity of 0.6. The level of Asn in LSCC tissue was higher than in adjacent tissue, with an AUC of 0.807, specificity of 0.737, and sensitivity of 0.882. This further confirmed that Asn played a pivotal role in the pathogenesis of tumors, may be a potential biomarker of LSCC, and may be used for the diagnosis or treatment of LSCC in the future.

The TNM staging system, an international staging standard established by AJCC and the Union for International Cancer Control (UICC), is an important clinical basis for treatment guidance and prognosis prediction of patients. The TNM staging system is classified on the basis of the primary tumor, regional lymph node involvement, and distant

metastasis. Early diagnosis and treatment of LSCC are crucial and can save manpower, financial resources, and time. In this article, amino acid metabolism detection was applied to the LSCC staging system in order to better determine the treatment and prognosis of LSCC patients. We found that Phe and Ile were the differential amino acids in the plasma, and Orn, Glu, and Gly were the differential amino acids in the tissue between stage I+II patients and stage III+IV patients. These amino acids had a good predictive effect on early (I and II) and advanced (III and IV) LSCC stages, as shown by ROC curve. Our study provided further insights into the early prognostic prediction of LSCC and the identification of novel targets or a molecular basis for the occurrence and development of LSCC.

Notably, our study was single-center with a small sample size. Future studies with larger clinical data are needed to verify our current findings, so that this tool can be used for the effective clinical diagnosis and staging of LSCC, bringing hope for the diagnosis and treatment of LSCC patients.

## CONCLUSION

In this article, the amino acids in the plasma and tissues of LSCC patients were detected using LC-MS, and Asn and Hcy were two amino acids of common significance in both plasma and tissue samples. In addition, Phe and Ile were two differential amino acids screened out in the plasma of LSCC patients at early (I and II) and advanced (III and IV) stages, and Orn, Glu, and Gly were three differential amino acids selected in the tissue. These dysregulated amino acids in LSCC patients may be promising potential clinical biomarkers for the early diagnosis and screening of LSCC.

### Funding
This work was supported by grants from the National Natural Science Foundation of China (81903094, 81600812), and the Natural Science Foundation of Henan Province (222300420566). The funders had no role in study design, data collection and analysis, decision to publish, or preparation of the manuscript.

### Grant Disclosures
The following grant information was disclosed by the authors:
National Natural Science Foundation of China: 81903094, 81600812.
Natural Science Foundation of Henan Province: 222300420566.

### Competing Interests
The authors declare that they have no competing interests.

### Author Contributions
- Shousen Hu performed the experiments, analyzed the data, prepared figures and/or tables, authored or reviewed drafts of the article, and approved the final draft.

- Chang Zhao performed the experiments, analyzed the data, prepared figures and/or tables, authored or reviewed drafts of the article, and approved the final draft.
- Zi'an Wang analyzed the data, authored or reviewed drafts of the article, and approved the final draft.
- Zeyun Li analyzed the data, authored or reviewed drafts of the article, and approved the final draft.
- Xiangzhen Kong conceived and designed the experiments, analyzed the data, authored or reviewed drafts of the article, and approved the final draft.

### Human Ethics

The following information was supplied relating to ethical approvals (*i.e.*, approving body and any reference numbers):

This study was approved by the Ethics Committee of the First Affiliated Hospital of Zhengzhou University (2021-KY-0417-002).

### Data Availability

The raw data are available in the Supplementaal Files.

### Supplemental Information

Supplemental information for this article can be found online at http://dx.doi.org/10.7717/peerj.15469#supplemental-information.

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
