# Peer review of "Clinical diagnostic value of amino acids in laryngeal squamous cell carcinomas"

_PeerJ, doi:10.7717/peerj.15469_

## Round 0.1 · original submission · Major Revisions

As stated by the reviewers, please address the following concerns and incorporate any necessary changes to the manuscript for resubmission.

Reviewer 1 ·

Basic reporting

This manuscript is a study discussing the use of using amino acid concentration as biomarkers for the clinical diagnosis of laryngeal squamous cell carcinomas (LSCC). The authors pointed out Asparagine and homocysteine were two amino acids of common significance in plasma and tissue samples, and their specificity and sensitivity analysis showed that they may be new
biomarkers for the diagnosis and treatment of LSCC. According to the TNM staging system, phenylalanine (Phe) and isoleucine (Ile) were screened out in the plasma of LSCC patients at early and advanced stages; ornithine hydrochloride (Orn), glutamic acid (Glu) and Glycine (Gly) were selected in the tissue. These dysregulated amino acids found in LSCC patients may be useful as clinical biomarkers for early diagnosis and screening of LSCC.

The authors used clear, unambiguous and professional language. The introduction contains background information for the readers. However, it would be better if the authors include more information about the current status of LSCC diagnosis, and which biomarkers are used currently. It would also be good if the authors can compare their discovery with current clinical practices.

The article was structured properly with sufficient data. Also the results were relevant to the data.

Experimental design

The experiments were designed properly. The research questions were well defined, relevant and meaningful. The methods were also described with sufficient detail.

Validity of the findings

The discovery from the authors can serve as a valuable information for further study and clinical diagnosis.

Reviewer 2 ·

Basic reporting

1) The paper will need to be revised one more time to make sure there is no typo and no ambiguous terms/sentences. For example, the author should put literature in the title on line 114 to the text below. In line 126, the first letter of analysis is capitalized which is not consistent with other subtitles. In line 141, please provide the full name of PLS-DA model and one or two corresponding references. On the first page, please add the full names for TNM classification and ROC. Although some terms are well known, it is better to mention the full name first in the article before using any acronym.

2) I would recommend doing one more round of searching on the published papers and adding them to the reference. In the statistical analysis section, it does not include papers that are directly related to the statistical methods and any clinical papers that have applied such methods which creates obstacles for the readers to understand the analysis approach.

3) Please provide the tables containing the VIP and p-values to match the results in Line 152 and Line 173.

4) All the Figures lack an explanation for the details which makes it difficult to understand. For example, in Figure 1a, what are PC1 and PC2? Please also explain the percentage values. If my understanding is correct, the PC is short for principal component and the percentage value represents the explained variance in proportion. However, not all readers can understand them without explanation. Besides, could the authors demonstrate why PC1 has a smaller explained variance than PC2 in Figures 5a and 5b? The permutations plot and heatmaps also need explanation. The texts inside the ROC plots in Figure 6 are so tiny. Please enlarge them.

Experimental design

1) The size of healthy volunteers is smaller than the LSCC patients which makes me confused. Since only fast blood sample is collected, it should not be difficult to involve suitable healthy individuals. Please demonstrate the reason why only 15 healthy individuals are involved.

2) The selected amino acids are based on some statistical methods. However, this paper lack details of statistical analysis. First, the authors should explain how you conducted adaptive scaling of data, what multivariate statistical analysis is used, and any software and packages used for each analysis part. I only see the package provided in the text about the heat map. The PLS-DA model should be introduced in the statistical analysis section. If any hyperparameter is applied, the authors should also mention that. Ideally, the analysis code should be attached to the supplementary material.

3) Please add more details about how you conducted the permutation test in Line 144.

Validity of the findings

1) The authors should demonstrate the reason that all the subjects in this experiment are male. Moreover, the authors also need to demonstrate the gender effect does not affect the result. Otherwise, the conclusion should be focused on male people.

Reviewer 3 ·

Basic reporting

This manuscript by Hu and coworkers titled ‘Clinical diagnostic value of amino acids in laryngeal squamous cell carcinomas’ describes the differences in amino acids concentration in 1) in plasma of laryngeal carcinoma patients and healthy individuals and 2) laryngeal cancer tissues and corresponding non-neoplastic normal tissue. Authors performed amino acids quantification in plasma of 20 laryngeal squamous cell carcinoma (LSCC) patients and 15 healthy volunteers. In addition, they analyzed tissues of all LSCC patients. This analysis throws light on a less explored concept in LSCC and I appreciate the authors for performing this study.
While LC-MS based amino acid profiling is the key element of this manuscript is very exciting, presenting this data as a biomarker study has severe limitations. The study design is not appropriate for the biomarker investigation. The main demerit in the study design is poorly defined research question and lack of appropriate controls. In a clinical setting, implementing the screening for LSCC requires defining a target population based on etiological characteristics such as individuals with history of smoking and alcohol consumption, oral HPV infection and gastrointestinal reflex disease. In addition, such diagnosis marker study should incorporate appropriate controls such as benign and precancerous lesions, other head and neck cancers. They limitations make this study weak.
I suggest authors to consider writing this manuscript as an observational study describing the amino acid profiles of plasma and tissues from patients and control. It would be nice to see how these amino acids vary in concentrations between these conditions and how much is concordance between tissue and plasma.

In addition, the following comments may help the authors improving the presentation.
1. Description of methods of bioinformatics analysis need improvement. Was the data normalized? What is the input data for the analysis? Is it abundance value or peak area? How were missing values handled? Did you use FDR?
2. Line 114: It is not appropriate cite literature in a subheading. Please explain it in the text.
3. Line 77: How this study helps in accurate staging of LSCC? You found that amino acids profiles differ between early and late stages of the disease. This does not mean amino acids will do better that gold standard pathological staging and they help in staging.
4. Why was study restricted to male subjects?
5. Authors used ‘metabolic profiling’ instead of amino acid profiling in several occasions (eg. Line 166)
6. Figure 3D: It seems Hcy has missing values more than half of the noncancer tissues analyzed. The difference shown in this figure could be due to higher frequency of missing data rather than real difference. How were missing values handles while data processing?
7. Is PLS-DA supervised and trained to find the differences in groups of comparison? What did Principal component analysis show? It would be interesting if PCA separates the groups as it is strictly unsupervised learning.

Experimental design

Please refer to basic reporting section.

Validity of the findings

Please refer to basic reporting section.

---

## Round 0.2 · accepted · Accept

All the reviewers' comments were addressed to the reviewers' satisfaction with the rebuttal submitted.

Reviewer 1 ·

Basic reporting

The authors have responded to the points in the previous review properly.

Experimental design

The authors have responded to the points in the previous review properly.

Validity of the findings

The authors have responded to the points in the previous review properly.

Reviewer 2 ·

Basic reporting

The authors have responded satisfactorily to all the comments and suggestions I made in my previous report.

Experimental design

No comments

Validity of the findings

No comments

Additional comments

No comments

Reviewer 3 ·

Basic reporting

My concerns were addressed. I have no further comments.

Experimental design

-

Validity of the findings

-